computational biology/biocomplexity/applied mathematics

pedestrian traffic, crowd dynamics, egress, dynamic scheduling, pedestrian sensing, real-time simulation

**Author for correspondence:**
Claudio Feliciani
e-mail: feliciani@g.ecc.u-tokyo.ac.jp

†These authors are joint first authors.

# A system for efficient egress scheduling during mass events and small-scale experimental demonstration

Hisashi Murakami[1,†], Claudio Feliciani[1,†],
Kenichiro Shimura[1] and Katsuhiro Nishinari[1,2]

[1]Research Center for Advanced Science and Technology, The University of Tokyo, 4-6-1 Komaba, Meguro-ku, Tokyo 153-8904, Japan
[2]Department of Aeronautics and Astronautics, Graduate School of Engineering, The University of Tokyo, 7-3-1 Hongo, Bunkyo-ku, Tokyo 113-8656, Japan

HM, 0000-0002-1433-0524; CF, 0000-0003-0718-8707

Improvements in the design of pedestrian facilities have reduced the frequency of crowd accidents, and safety is now generally ensured in well-planned crowd events. However, congestion and inefficient use of infrastructures still remain an issue. To guarantee comfort and reduce close contacts between people, there are circumstances when crowd density may have to be reduced well below safety limits. Although research has given a lot of attention to extreme scenarios, methods to improve non-critical conditions have been little explored. In addition, crowd sensing technology is still mostly used for data collection and direct use on crowd management is rare. In this work, we present a system aimed at computing optimal egress time for groups of people leaving a complex facility. We show that, if egress starting time is accurately computed for each group based on actual crowd conditions, density can be greatly reduced without having a large effect on the total egress time of the whole crowd. To show the efficacy of such a system, a small-scale experiment is conducted where all components are tested in a simple scenario. As a result, an increase in total egress time by only 5% allowed to reduce maximum density by 35%.

## 1. Introduction

A growing number of countries worldwide are faced with the challenge of controlling crowd movements and ensuring no accident or dangerous situations occur while people are moving from place to place or are attending a large event or a

demonstration. As a result of these social changes, an increasing number of researchers are focusing their attention on the topics of pedestrian traffic and crowd management. An initial interest in qualitative features from the start of the twentieth century [1,2] (mostly related with mass psychology) has led to the definition of various crowd theories [2–5]. Later on, a shift to quantitative aspects has been observed around the 1960s [6], when cities started to grow more rapidly, air-travel became more affordable and data on mobility became available in large number.

Volume of research related with quantitative crowd dynamics and pedestrian traffic, the central topic of this work, has seen a clear connection with available technologies. In the early time, pedestrian movements had to be measured using stop-watches or in a static way from pictures taken at regular intervals [6,7]. With the emergence of low-cost camcorders in the 1980/1990s, obtaining recordings of pedestrian spaces to be later analysed numerically became easier, and quantities such as speed or flow got measured with an increasingly higher accuracy [8,9]. Later, the emergence of high-performance computing allowed the direct analysis of videos using dedicated algorithms, thus allowing to track pedestrians under specific conditions and compute more detailed quantities such as pedestrian density [10]. Modern pedestrian tracking systems employ different technologies and are available in large number. In addition, those systems often operate in real-time, making the collection of crowd data a relatively easy task [11,12]. However, the large availability of data in regard to crowd dynamics and the real-time novelty did not significantly translate into a change in the use of those data. Pedestrian data (often trajectories) are typically either collected for later analysis [13,14] or used to calibrate simulation models [15–17].

On the other hand, also the aim and motivation for research in crowd dynamics have not changed a lot despite the social/technological context having seen a constant transformation throughout the twentieth and twenty-first centuries. A great deal of research on pedestrian crowds is still focusing on emergency evacuations or extreme conditions. A common motivation to justify this particular attention is given by the fact that stampedes or crowd accidents, while rare, can lead to several people getting injured and, in reality, several deaths are reported every year around the world [18,19]. Although there is no doubt that understanding the reason for past crowd accidents and preventing their occurrence in the future is of utmost importance, it should be also partially acknowledged that most of the accidents occur due to quite recurring organizational failures [20–23] (missing coordination between stakeholders, planning deficiencies, no control over ticket issued, etc.). Under this perspective, it could be argued that rather than gaining a new understanding on crowd dynamics, having the accumulated know-how applied when planning crowd events should be the priority, if accidents are to be prevented. In this regard, it can be generally observed that structures designed to accommodate a large crowd on a regular basis (like stadia or exposition halls) rarely register crowd accidents, due to the fact that design is carefully studied and staff well trained and available in sufficient number.

In brief, experience gained from tragedies that occurred in the past, strict regulations imposed on operators of pedestrian facilities, and constantly improved simulation models used to design them, all contributed in reducing the number of crowd-related casualties. In this context, safety in case of accidents (e.g. fire) is typically guaranteed in modern and well-designed facilities, thus allowing operators to focus on crowd management during normal operations and set attention on occupants' well-being and minimize risks to which they are potentially exposed. However, it should be also noted that not only external triggers (e.g. fire, power outage, earthquake) could undermine occupants' safety, but the crowd itself could constitute a risk which has to be managed by facility operators. For example, keeping a division between football fans belonging to opposite teams can help to reduce the risk of violence. Also, there are situations during which it is necessary to avoid a mixture between different groups of people for health reasons (like in the COVID-19 pandemic crisis). Finally, when evacuation safety is ensured by proper practices and other risks are minimized, the priority of facility operators is typically moved to comfort-related issues, specifically focusing on improving visitors' overall experience by reducing congestion and crowded areas.

Due to the improvements in evacuation safety, these latter non-emergency-related aspects of crowd management are playing an increasingly important role and require a different approach compared with the classical strategies used to prevent accidents during evacuation. When crowd behaviour during normal conditions is concerned, worst-case scenarios are not anymore the reference and time is not the main aspect, but the actual crowd condition represents the starting point to prepare crowd control strategies based on multiple variables. As such, information gained from crowd sensing technologies represents the basis to prepare optimal strategies aimed at further reducing crowd intrinsic risks and improving the overall experience of pedestrian users. However, in spite of the increased importance attributed to real-time pedestrian data and the need to elaborate strategies based

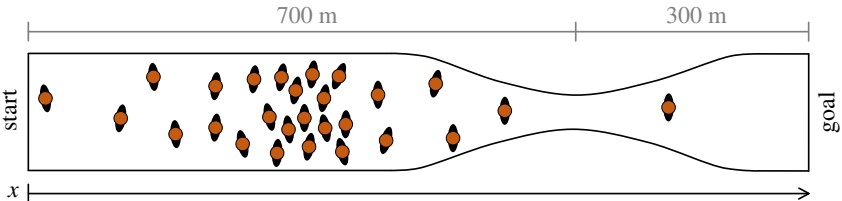

**Figure 1.** Layout for the 1 km race considered in this introductory example. We will assume that a restriction is present 700 m after the start and two runners at most can pass simultaneously. Runners are schematically represented over the course in a small number.

on actual conditions, research on pedestrian dynamics has given little attention to this emerging reality. Although some studies did consider the use of real-time pedestrian positions to optimize evacuations in case of accidents [24–26], to the knowledge of the authors, only a few attempted to reduce risks and/or improve pedestrian comfort during normal operation [27].

In this work, we present a real-time system making use of pedestrians sensors to estimate the optimal egress time in complex crowd facilities in situations where a large number of people need to leave a facility almost simultaneously. In our framework, a balance between time and 'experienced crowd density' is considered to reduce risks and improve pedestrian overall experience under normal operation. As we will see, it is possible to ensure a safe and uncongested egress, while ensuring that groups from different areas do not mix, without a consistent increase in the total egress time. The framework presented here should be seen as a method to ensure occupants' comfort during normal operation and could be employed in addition to alternative approaches aimed at optimizing evacuation time in emergencies [24–26] and for which their contribution becomes significant only or mainly in case of accidents.

This work is organized as follows. In the next section, the conceptual and theoretical background related with this work is presented and discussed. Next, technical aspects are introduced, namely the simulation model and the algorithm computing optimal egress time by employing a sensing system to detect pedestrian position. Results relative to the system are later presented and conclusions are drawn at the end including an exhaustive discussion on limitations and potential future studies.

# 2. Theoretical and conceptual background

To explain the underlying concepts of this work and its potential applications, we will consider different scenarios starting from a simple one to increasingly complex ones. Although each scenario is different, the underlying principle will remain the same and the only difference consists of technological requirements and mathematical complexity needed to deal with each scenario.

## 2.1. Running competition

Let us first start with a very simple case, whose solution can be computed mathematically using elementary algebra. We will assume that a 1 km athletic race has been organized in an old town and runners will have to pass through streets of different widths. A schematic representation of the course is given in figure 1. As given in the diagram, although most of the race will run through wide roads, a restriction is present 300 m before the arrival (i.e. 700 m after the start) and organizers are concerned that congestion will form at the restriction and runners will not be able to compete at their best. It is estimated that at most two people can pass simultaneously in the narrow section.[1]

Although the race is supposed to start in block with all participants starting together, organizers understand that this will lead to problems at the narrow section and therefore runners will have to be divided into groups. For the sake of simplicity let us assume that runners (1000 in total) are equally divided into half running at a speed of $3.0 \pm 0.5 \, \mathrm{m \, s^{-1}}$ and another half running at $4.0 \pm 0.75 \, \mathrm{m \, s^{-1}}$. To ensure a good level of competitiveness, organizers are therefore considering to delay the start of the fast runners (the 500 people running faster) and make sure that congestion does not build up in the narrow section.

The minimum delay required to ensure that congestion will not form in the narrow section can be computed considering the distribution of the runners over the course. For the sake of simplicity, we

---

[1]For simplicity, we will consider a unidimensional case here and density is intended as people per metre course, similarly to the cars per km used in vehicular traffic.

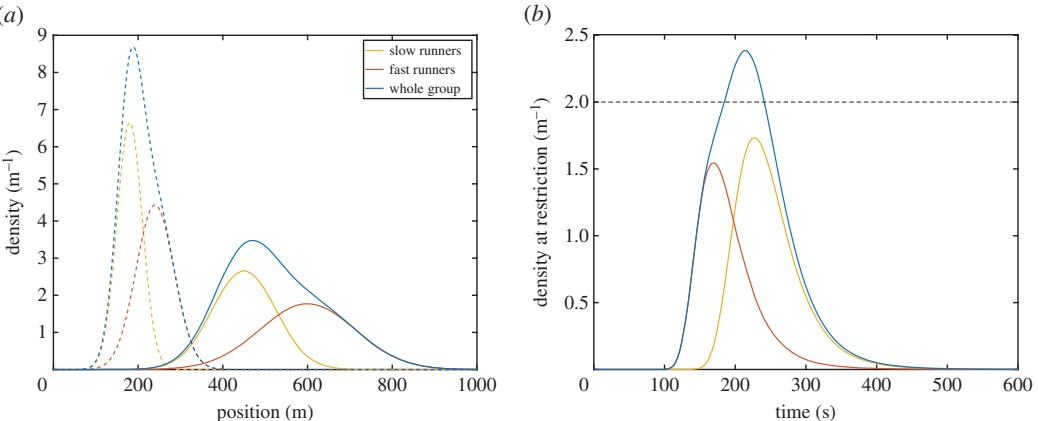

**Figure 2.** (a) Distribution of runners at two distinct times over the course. Selected times are 60 s (dashed line) and 150 s (solid line). Density refers to the number of runners per metre course, similarly to the definition used for cars in vehicular traffic (independently on the width). (b) Density at the restriction ($x = 700$ m) as a function of time. The contribution of each group to the total density is shown and the horizontal line indicates the maximum density allowed to ensure a smooth race.

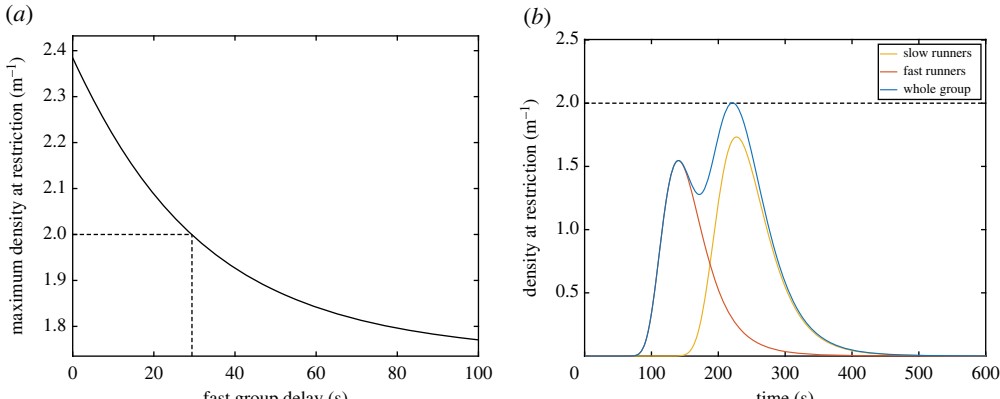

**Figure 3.** (a) Maximum density reached at the restriction in function of the delay imposed on fast runners. (b) Temporal evolution of density at the restriction when the optimal delay is imposed. Maximum density reaches exactly 2 m$^{-1}$.

will assume that all runners start simultaneously from the position $x = 0$ m (this is, however, normally not possible for a large number of participants). Since pedestrian walking speed and running pace are normally distributed [7,28], the distribution of runners over the course will decrease over time in a sort of diffusion–advection process.[2]

After a time $t$, the centre of each group will be at the position $\mu = v_{avg} \cdot t$ ($v_{avg}$ being the average speed of the slow/fast group) and the distribution of their positions will have a standard deviation of $\sigma = v_{std} \cdot t$ (with $v_{std}$ being the standard deviation for the group speed). Figure 2a presents the distribution of the slow and fast runners along with the distribution of the whole group for two selected times (in the case of a simultaneous start). The functions of both space-distributions can be used to compute the time-dependent density at the restriction as shown in figure 2b. As seen, with a simultaneous start, the overall density considering both groups exceeds the maximum of 2 m$^{-1}$.

By shifting the time-density curve relative to the fast runners to the left (thus delaying their start) it is possible to reduce the maximum density at the restriction. Figure 3a shows the maximum density reached at the restriction in relation to the start delay imposed on fast runners. It can be found that a delay of 29.4 s will result in a maximum density of exactly 2 m$^{-1}$ and thus any further delay will further reduce the density. Figure 3b shows the time-density relationship using the minimum delay; as it can be seen, the maximum density reaches the value of 2 m$^{-1}$.

---

[2]Using the same advection–diffusion theory also the case for a 'group' start (in which runners are distributed over a distance) could be considered, but the mathematical treatment is more difficult as the implicit error function is involved [29].

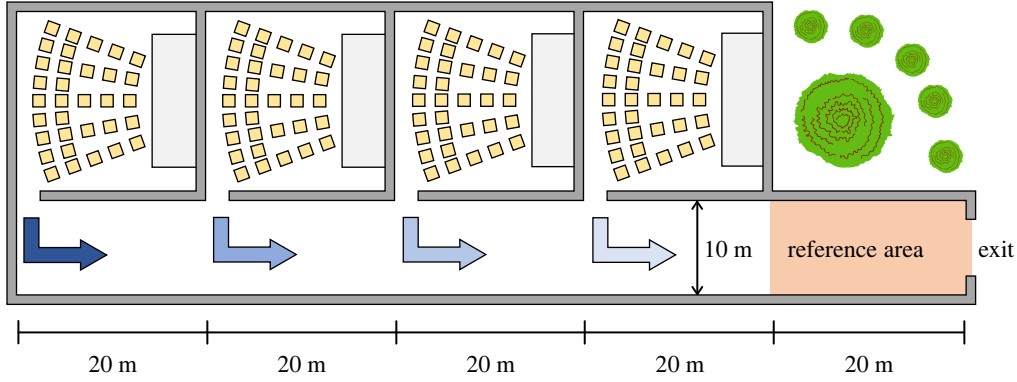

**Figure 4.** Schematic for a conference centre or a multiplex cinema complex. Each room is connected to the exit through a long corridor. The area used to compute in- and outflow and the density is the 'reference area' given in red.

As explained in this example, a short delay imposed on a selected group of people would be sufficient to ensure a group arrival and keep the level of competition high, while, at the same time, avoiding problems related to congestion.

## 2.2. Conference venue

In the previous example, we considered a simple case, which could be solved even analytically to obtain the exact solution of the problem. Although we tried to consider a scenario close to reality, several simplifications and assumptions had to be done. Here, we wish to consider a partially more difficult problem, which would help to get closer to the goal scenario motivating this study. Also, the main analytical tool of this work will be presented while discussing this example.

In this example, we will consider a building with four rooms, such as a conference centre or a multiplex cinema like the one represented in figure 4. The four rooms are all connected to a long corridor giving access to the exit. Each room is 20 m long and the corridor is 10 m wide. In this case, we will assume that the operating company wants to ensure that events in the single rooms are scheduled with a sufficient delay to guarantee that people would never leave simultaneously from each room (in the case of cinemas start and length of each movie can be precisely known and controlled by eventually extending half-time break or adding commercials). Since each event will attract a different number of people which is not always possible to predict, the managing company is interested in a system which would dynamically compute optimal delay based on real-time conditions.

Although even this problem could still be possibly solved analytically, we are not interested in the exact solution, but we would rather show an approach allowing us to solve this problem in a simple way, also considering the technology available nowadays. In this case, we will apply a Monte Carlo method and perform simple simulations to show the differences between different strategies and how to find the optimal delay between each room.

To perform the calculations, we will assume that each room is filled with 350 people and the *average* flow at door is 3.0 persons s$^{-1}$ (we will assume that people will not get out continuously but in a rather dispersed way, which is the case if there are groups and no urgency). We will further assume a typical distribution for the walking speed of $1.5 \pm 0.25$ m s$^{-1}$. As we will see later, to compute the density in the reference area, the (in)flow on the left side and the (out)flow on the right side are needed. To compute these flows, it is sufficient to know the number of people reaching each point at any given moment. Since we are assuming that people randomly leave the door of each room after the event is finished, we may assign a randomly selected start time using the average flow and the number of occupants. Thus, each person will be assigned with a start time uniformly distributed between 0 and 116.7 s from the 'opening of the door'. Using start time, walking speed and the geometric dimensions, the number of people entering the reference area and leaving the exit can be computed by summing up the people being there at a specific time interval (one second here).

The so-called in- and outflow computed using this Monte Carlo approach are shown in figure 5. As shown in figure 5, three scenarios are considered: a simultaneous egress from the three rooms, a fairly delayed egress (balanced delay) and an overly delayed egress (excessive delay, being exactly double the balanced delay). To compute each scenario 10 000 repetitions have been performed and the average trend has been used for further calculations.

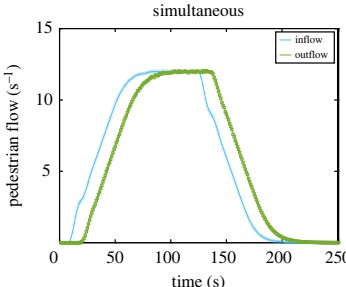 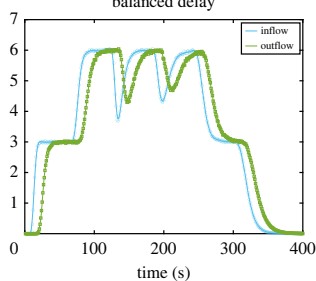 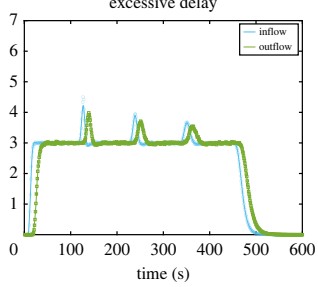

**Figure 5.** In- and outflow in the reference area computed using a Monte Carlo method and fitted using a smoothing spline. In the graph on the left people leave simultaneously from the four rooms. In the case in the middle a delay of 150 s, 100 s, 50 s and 0 s is set in each room, with the delay reducing the closer the room is to the exit. In the case on the right, a double large delay is set for all rooms. In all the cases, the restriction relating to the maximum flow on the exit is not imposed yet.

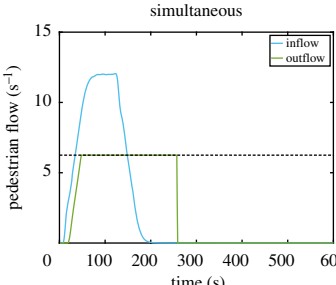 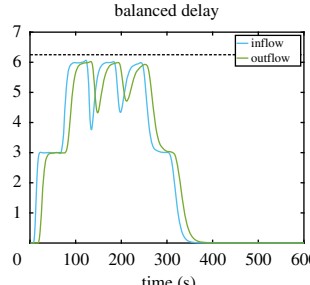 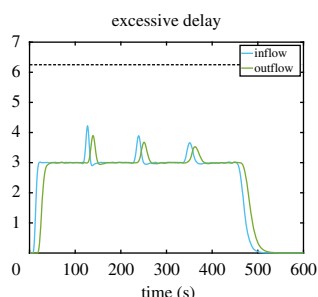

**Figure 6.** In- and outflow in the reference area after imposing the maximum flow condition at the exit. Fittings obtained from the results given in figure 5 are used. The order of the figure is the same as in the previous graphs.

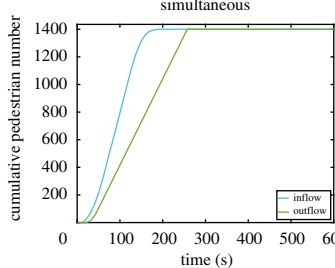 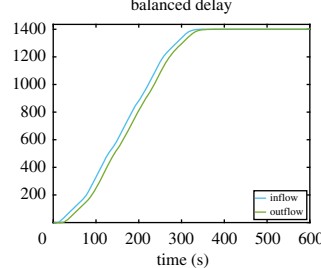 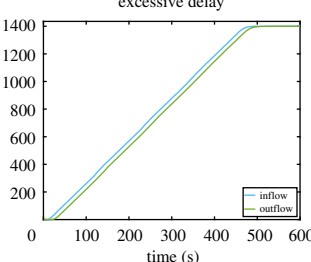

**Figure 7.** Cumulative in- and outflow for the three set of delays considered. The total number of occupants (1400 people) is reached at the end of the egress.

The in- and outflow presented in figure 5 would represent the case of an open exit, where no restrictions are imposed on the maximum flow. However, we will assume, as it is often the case, that the exit has a maximum flow, having a value of 6.25 persons s$^{-1}$ in this example. This means that if the outflow exceeds the maximum limit, people will start accumulating in front of the exit and the outflow will keep its maximum value until few people are left in the corridor. Figure 6 shows the in- and outflow in the reference area after imposing the maximum flow condition at the exit. As it can be seen, when no delay is imposed there is a large imbalance between the in- and outflow, meaning that people are accumulating in front of the exit. On the other side, when delays are used, the outflow is simply a sort of translation of the inflow as people are able to freely walk through the exit unimpeded. However, if delays are not accurately chosen, waiting times may unnecessarily grow, as seen for the case on the right.

To analyse this example a little further, we may also need to consider the density in the reference area. Density can be easily computed by counting the number of people within the reference area at any given time. This can be obtained using the cumulative in- and outflow, where the difference between both curves provides the number of people there at any given moment. Cumulative in- and outflow for the three cases considered are given in figure 7.

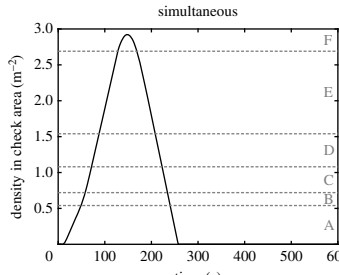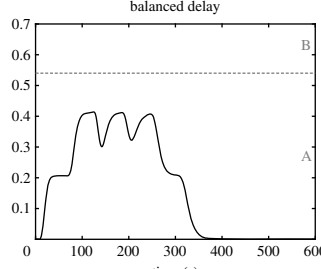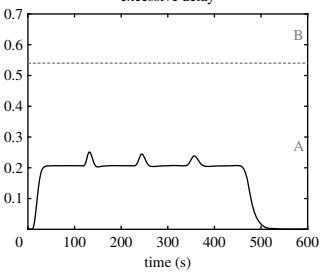

**Figure 8.** Density in the reference area for the three scenarios considered. Horizontal lines and letters on the right indicate the level of service as defined by Fruin for flat walkways [6]. The scale A to F represents the 'comfort' from the perspective of pedestrian users, where values above F are considered unacceptable in the design of pedestrian facilities.

**Table 1.** Egress time, maximum density and cumulative product (a measure for overall efficiency) for the three delay combinations. Egress time is defined as the time from the start (time 0 s) until the moment when 10 people are left in the building (extremely slow people may affect the results, so a tail is used). Room 1 is the one closest to the exit, 4 the furthest.

| condition | door opening delay (s) | | | | egress time (s) | maximum density ($m^{-2}$) | cumulative product (s $m^{-2}$) |
|---|---|---|---|---|---|---|---|
| | 1 | 2 | 3 | 4 | | | |
| no delay | 0 | 0 | 0 | 0 | 255 | 2.92 | 94 940 |
| balanced delay | 0 | 50 | 100 | 150 | 337 | 0.41 | 32 551 |
| excessive delay | 0 | 100 | 200 | 300 | 487 | 0.25 | 47 041 |

The density in the reference area is shown in figure 8 for all the combinations of delay considered. To help reading the results, the level of service (LOS) [6], which determines the 'comfort' of pedestrian facilities, is provided along the density curves. In the case without delay, a density of almost three people $m^{-2}$ is reached. While this would be no concern for an evacuation, and it is not risky in terms of safety, we would like to remind that in this work we are not focusing on evacuation, but service quality and pedestrian comfort in normal conditions. In this regard an LOS F relates to an unacceptable design and a facility where congestion is frequent would hinder the satisfaction of users. On the other side, both cases with a delay are well below the lowest LOS and would therefore represent the ideal case for pedestrian users.

Even if density is a concern for pedestrian users, time is also important to them. In this regard, to conclude this example, we wish to compare the three cases in a more systematic way. Table 1 presents the total egress time and maximum density for the three scenarios considered. As it can be seen, the case without delay is overall the fastest, but its efficiency in terms of time is compensated by high densities. On the other side, if delays are not accurately chosen, egress time may grow exceedingly large without resulting in a significant reduction in density. The case with an excessive delay has basically the same density as the case with a balanced delay (at least in terms of LOS), but is 45% slower. However, when the delay is balanced, egress time is increased by 'only' 32%, but density falls by more than 85%. If a conference venue or a multiplex cinema is considered as in this example, imposing a delay of 2.5 min would not result in substantial issues related to the daily schedule but could greatly improve user experience.

It can be concluded that the optimal delay combination should ensure a trade-off between the increase in egress time and the reduction of density. This condition can be simply summarized by stating that the product between egress time and density should be minimized to obtain the optimal delay. In the specific, in this study, we found that the optimal delay is more easily found by minimizing the product (which we will call 'cumulative product') of the egress time and the cumulative density, which is defined as the sum of all densities recorded in the reference area throughout the whole egress process (interval for 'density sampling' is defined as one second here). To justify this, we noticed that the use of the cumulative density allows imposing a further penalty in relation to egress time and helps in defining more clearly the optimal delay.

When the above definition is applied to the three scenarios considered in this example, it is clearly seen that the cumulative product is the lowest in the case with a balanced delay. Although in this

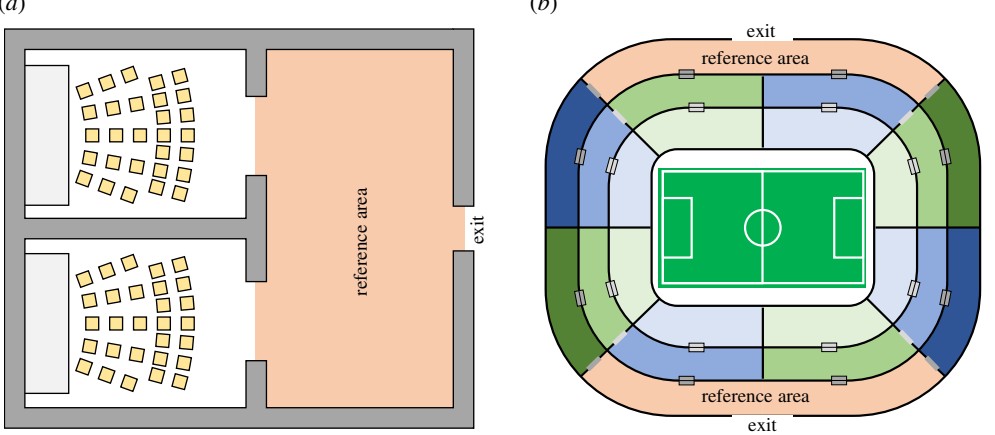

**Figure 9.** (*a*) Case study considered in this work. This is basically a simplification of the conference centre/multiplex cinema considered earlier. (*b*) Example for a possible application of the concept presented in this work. Although the structure is much different, the analytical approach would remain unchanged.

example, we simply introduced the conceptual framework, in the next example we will show that optimal delay can be actually computed by using the definition of the cumulative product.

## 2.3. The double room case

From the previous examples, it is clear that when both the perspectives of facility operator and its users are considered, then, taking a trade-off between egress time and density at the exit seems the optimal solution. As a consequence, the optimal delay for each section of a facility may be found as the solution of a minimization problem, where the minimum of the product by egress time and cumulative density is searched. While in the race case the optimal delay could be easily computed, in the previous example a three-dimensional minimum problem would have to be solved. Although this is not mathematically difficult (looking for a minimum is no different to calibrating a simulation model and this is often done in pedestrian or traffic simulation [30–32]), it nonetheless requires computational power and would be mostly a challenge from a technical point of view.

Considering the reasons listed above, in this study we will focus on a simple case to demonstrate the validity of the proposed approach and show how it would be possible to actually build a system to schedule egress time based on real-time conditions. For instance, we will consider a simplification of the previous scenario, where only two rooms are present as shown in figure 9a. This is much easier from a mathematical point of view (only one parameter, e.g. a single delay, need to be computed) and it could be tested in real conditions using real people and a system implementing both sensing and egress scheduling.

With this said, we want to point out that the same approach used in the two-rooms case (and discussed earlier for the four-rooms equivalent) could be used in much more complex structures. Let us consider for example a soccer stadium such as the one schematically shown in figure 9b. Most modern stadia have a limited number of exits used for normal operations (in addition to emergency exits which may be opened in case of accidents) and different areas separating fans and people holding different classes of tickets. If layout of the stadium is known along with the number of people in each section, egress simulations can be conducted in a short time using modern simulation models and computing power [26,33–35]. Also, for modern facilities, attendance can be estimated well in advance based on ticket sales. This means that optimal egress time could be roughly estimated in advance based on ticket sales and later further optimized using real-time data, thus reducing the need for computing power.

# 3. Simulation model and numerical analysis

From here on we will focus on the two-rooms case study, whose solution regarding the optimal egress time will be presented in this section. Here, we would like to remind the readers, that, while the scenario is fairly simple and a solution may be also found analytically with some simplifications, our

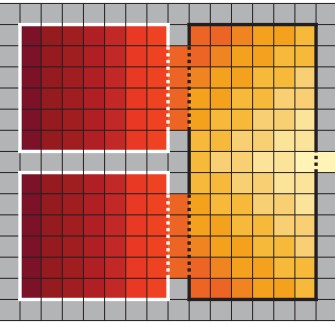

**Figure 10.** Static floor field in the double room scenario. Darker colours stand for higher values of the static FF, with the exit shown using the lightest colour. Starting rooms are given on the left half (and divided by a wall) and the reference area on the right half (with the exit in the middle of the right wall). Starting rooms are shown using white lines, the reference area in black. The whole room (including walls) is $16 \times 15$ cells in size.

goal is to test a scalable system which may be adapted to larger and more complex structures (like the stadium discussed above) without changing the underlying analytical approach.

## 3.1. Simulation model

In this study, we adopted a standard model of pedestrian dynamics called the floor field (FF) model [36–38]. In this model, the motion of each individual is represented in discrete time and space and implemented through computational time steps and using a mesh where a maximum of one individual per cell is allowed. At every time step, each individual selects one of its four adjacent cells or decides to remain on its current site. The basic concept of the FF model suggests that individuals move according to two kinds of FFs: the static FF which guides them to the destination and the dynamic FF which allows to create emergent structures (lane formation, oscillations at bottlenecks, etc.). However, because the dynamic FF does not impact simple evacuation situations [39,40], we will ignore it here. We therefore employed only the static FF that allows pedestrians to find the shortest path to the exit. The static FF employed for the double room scenario is very simple (figure 10), and it simply represents the minimum distance (in cells) to the exit if a Neumann neighbourhood is employed (i.e. in the model people cannot move diagonally but only along the four directions). Under this condition, the value of the static FF decreases from the back of the rooms (where it equals 20 'steps') until reaching its minimum (zero) at the exit.

Employing the static FF from figure 10, the transition probability $p_{i,j}$ allowing each individual to select the target cell during each time step is computed on each neighbour site $(i, j)$ according to:

$$p_{i,j} = N \cdot e^{K_s S_{i,j}} \cdot \xi_{i,j}, \tag{3.1}$$

where $S_{i,j}$ is the local static FF value, $K_s$ its sensitivity parameter (set to $K_s = 3.0$) and $N$ is the normalization coefficient (setting the sum of the probabilities for all neighbouring sites equal to 1). $\xi$ is the obstacle parameter: returning 0 for walls or occupied sites and 1 for empty cells. Individuals compute the transition probability for all neighbours and move to the one having the highest probability. However, when several individuals attempt to move toward the same site simultaneously a conflict would occur. In this case, a randomly selected individual will be selected among the contestants and allowed to move there (with the remaining ones having to stay in their position).

In this study, we set the cell size at 60 cm and the time step interval at 0.33 s. It should be noted that cell size used in this study is somewhat larger than typical values used in floor field simulations (typically around 40 cm [7]), but reflects the fact that in validation experiments (presented later) competition was low and density never exceeded 3–4 people m$^{-2}$. Also, cell size was taken to correspond with the waiting positions (presented in detail in the next section), meaning that each of the $7 \times 6$ cells in both rooms (given in white in figure 10) could be occupied by a pedestrian before the start of the experiments. Exit width was also chosen in accordance with experimental settings.

## 3.2. Numerical analysis

Optimal egress delay was computed using the simulation model presented above and the method discussed earlier for a situation in which 30 individuals are initially present in both rooms (and

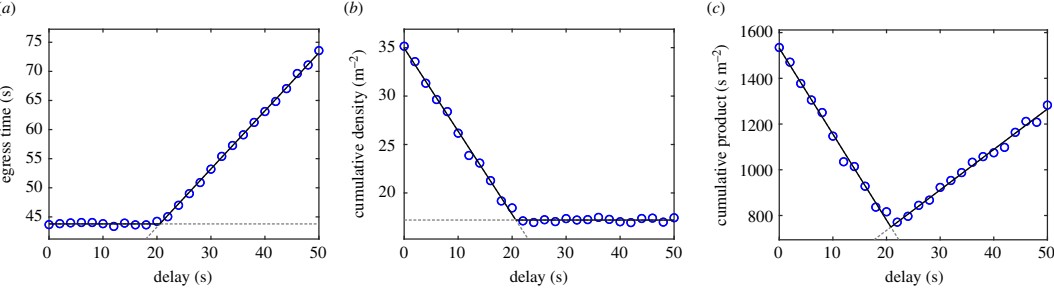

**Figure 11.** Egress time (*a*), cumulative (or experienced) density (*b*) and the product of both factors (*c*) in relation to the imposed delay. The cumulative product can be used to estimate the optimal delay by taking its minimum.

aligned to occupy all positions facing the exit, thus leaving the last two rows empty). Delay between the first and the second group (either the lower or upper room, considering the geometry is symmetrical) was changed from 0 to 50 s in steps of 2 s and 10 simulations were conducted for each condition.[3]

In line with the examples discussed earlier, egress time, cumulative density in the reference area (the black region in figure 10) and the product of both quantities were considered. Since simulation time step was smaller than a second, cumulative density was normalized as the sum of the density recorded in a second interval through the whole egress process. Egress time was taken using the whole group of 60 participants. Simulation results for the three quantities considered are shown in figure 11.

In figure 11, it is clearly seen that egress time remains fairly constant until 20 s, after which it starts growing linearly. The increase in egress time is given by the fact that the reference area remains empty for a short time after the first group leave, thus resulting in an inefficient egress which gets the more and more inefficient when larger delay is taken. However, for the same reason, cumulative density would not increase after around 20 s of delay. Since group motion is completely separated for large delays and the reference room is empty between both groups' start, cumulative density would not increase despite longer overall egress times. By contrast, simultaneous start results in a short egress time, but density remains high all the time.

When the cumulative product is used, the optimal delay is simply found by finding its minimum. As it can be seen from figure 11 a 'V'-shaped graph is formed and the optimal delay can be more accurately computed by linearly fitting both sides of the 'V' and taking the intersection between both curves. This results in an optimal delay of 20.68 s.

# 4. Experimental set-up and methods

To validate the approach discussed in the previous section and test its capability in a real situation, the double room condition has been experimentally recreated and investigated using recruited participants. The system described in this section may be conceived as a small-scale representation of a larger infrastructure, like a stadium or the multi-purpose conference centre discussed earlier. In the following, we are going to describe aspects related to the experimental investigation, including both procedural and technical details.

## 4.1. Experimental design and procedure

The double room scenario has been recreated in a large lecture hall (which partially explains the restrictions imposed on size and number of participants relative to the previous section). A schematic of the experimental design is provided in figure 12, while figure 13 presents a frame from the main camera (above the exit) and a view from the ground showing an empty waiting area from the perspective of participants.

Two waiting areas were created next to each other and in each area waiting positions were drawn on the ground as shown in figure 12. Seven rows comprising six positions each were created in each room, with a distance between each neighbouring row and column set at 60 cm. The given configuration (equal to the one used in simulations) allowed therefore to accommodate a maximum of 42 people in each room.

---

[3]Number of repetitions is relatively low, but was sufficient to get accurate results and was also chosen considering the real-time approach discussed in the next section while presenting the system.

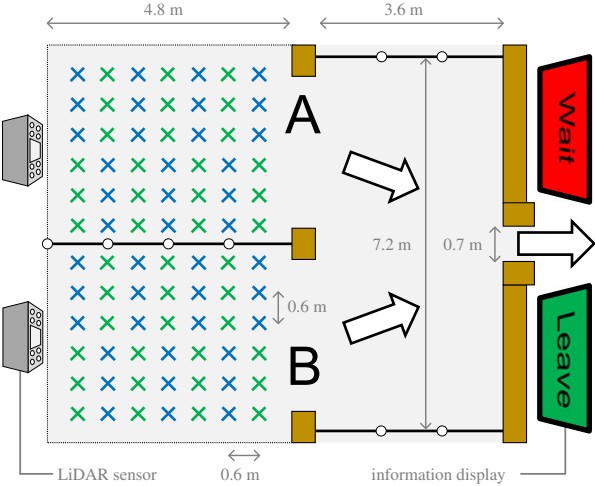

**Figure 12.** Geometrical configuration of the experimental area. In the example provided here participants in area B (below) were allowed to leave the room before the ones in area A (above). Position of both LiDAR sensors is also schematically given.

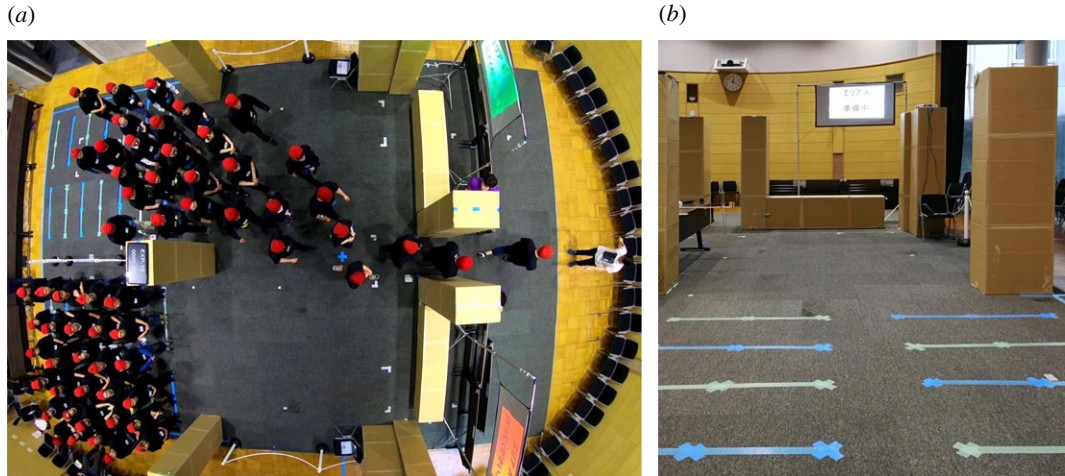

**Figure 13.** (*a*) Top view of an experimental trial taken using the camera also employed for trajectory extraction. Starting positions drawn on the ground and timing screens are clearly seen and a member of staff is also shown on the right side guiding participants on both sides after passing through the exit. (*b*) View from waiting area B during the preparation of the experiments. Note: the desk partially seen on the left in the middle of the reference area was removed during the experiments.

A buffer room (or reference area, using the terminology used so far) 3.6 × 7.2 m in size was dividing the waiting areas and a 70 cm exit (seen on the right of figure 12 and figure 13*a*) and participants were asked to move through the exit after the start of each experiment. The reference area was delimited using cardboard boxes and chain partitions.

External sides of the waiting areas were open and participants could enter them from the outside. However, the two waiting areas were separated by a chain partition and participants were not allowed to move to the neighbouring waiting area after the start of the experiment.[4] Each experiment started with an acoustic countdown (lasting 5 s), but, depending on the condition, only participants in one room or in both were allowed to walk toward the exit and leave the experimental area. Two large screens were used to indicate if and when participants in each room were allowed to walk toward the exit. The screens were set at a sufficient height so that all participants (even the ones on the back) could see them.

---

[4]Due to a mistake during experiment preparation, a slightly larger exit was used in the waiting areas, e.g. one cardboard column was used instead of the planned three (compare figure 10 with figure 12). However, we believe this mistake had little influence on the results.

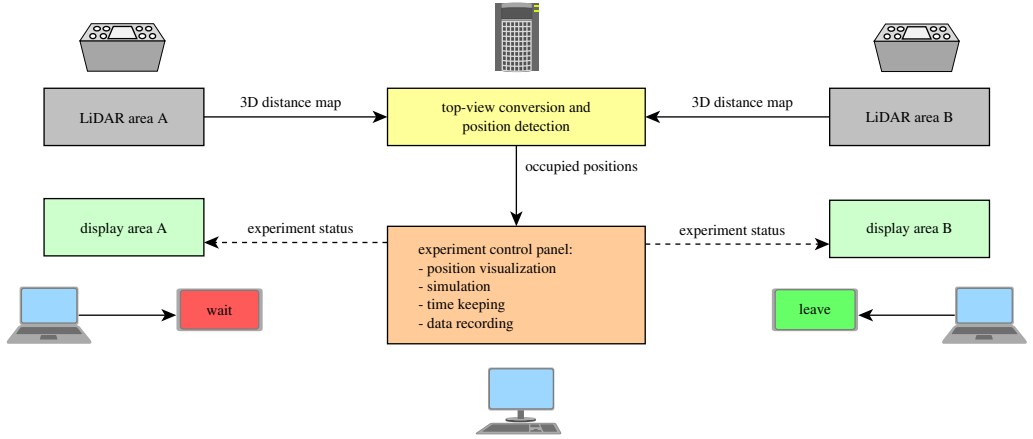

**Figure 14.** Experimental technical set-up showing all components and their connection. Solid lines indicated cable connections, dashed lines wireless connections.

Before each experiment, when participants were moving to their positions, each screen showed an 'in preparation' message on a white background. When the 5 s long countdown started, both screens turned red and a 'wait' message was shown on both sides. When participants were allowed to move a 'leave' message was shown on a green background. Timing including the countdown, delay between both areas' start, leaving order and messages displayed on both screens could be controlled through a control panel presented later on.

After transiting through the exit, participants walked back to their respective waiting area passing to the external sides of the experimental ground. A member of staff was placed outside the exit to guide participants on both sides and avoid that the region past the exit would get congested. Participants were instructed to walk normally and avoid rushing or unnecessarily wait in their positions when allowed to move. Participants were not informed on experiment's goal or motivation and simply asked to walk through the exit once the 'leave' message was shown.

## 4.2. Technical set-up

To compute optimal egress delay while considering positions occupied by pedestrians in each room and the corresponding number of participants, a computerized system connecting sensing equipment with the displays was developed for the scope of this experiment. The overall system is graphically sketched in figure 14 with each component and their connections explained in detail hereafter.

Two time of flight (TOF) light detection and ranging (LiDAR) sensors (manufactured by Hitachi-LG, product number HLS-LFOM1) were installed in each waiting room to detect participants' positions and eventually count their number (both sensors are schematically represented in figure 12). Sensors were installed 125 cm behind the last row of starting positions, centrally located in the width of the waiting area and set at a height of 300 cm (using a tripod). Sensors were regulated to an angle of 55° toward the ground (0 being the horizontal direction). Since the sensors employed have a maximum detecting distance of 10 m and a resolution of $320 \times 240$ pixels, both areas could be inspected with enough accuracy to detect the presence of participants on each waiting position.

Software was developed to convert distance raw data obtained from the LiDAR sensors to a top-view map of each waiting area showing distance of objects from the sensor and their height relative to the ground. Figure 15 provides two examples of frames showing top-view converted distance data in which an interface specifically designed for this experiment is also included to detect positions (details following).

To detect whether participants were actually occupying a given position in the waiting area or not, a simple algorithm was developed. First of all, both sensors were arranged and calibrated to make sure that the coordinate system within the sensing algorithm would correspond with the planar axis in the waiting areas. After obtaining a sufficient accuracy, waiting positions were defined in the sensing system according to the experimental geometry. For each position, an area having a radius of 0.1 m and centred on the cross mark on the ground was virtually defined within the sensing system. People would get detected as being in that position if the maximum height in that area would exceed 0.95 m

detected positions in area A          detected positions in area B

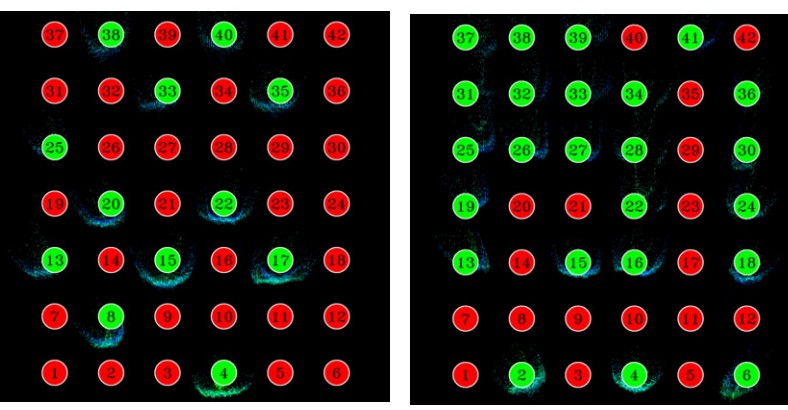

**Figure 15.** Detected participants' waiting positions based on distance data gained from sensors and converted into top-view heights points. Small dots indicate a detected object with its colour indicating the height from the ground (a 'temperature' colourmap is used, blue being the lowest, red being the highest); body shape is clearly recognized for people close to the sensors. In this case, both participants' number and positions are correctly detected. Note that positions are rotated 90° respect to the representation of figure 12.

(a)                                    (b)

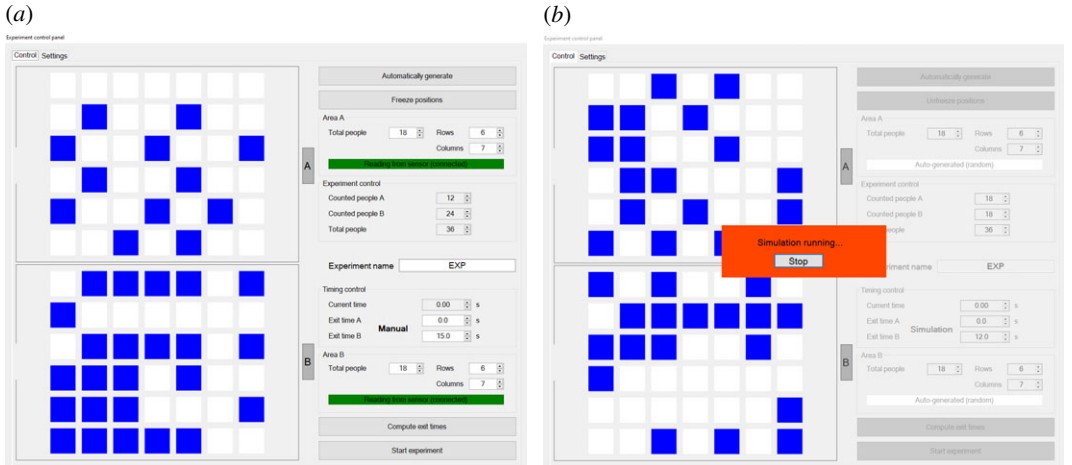

**Figure 16.** (a) Control panel showing the waiting position on both areas (the same as figure 15, rotated another 90° here) and providing a control over the experimental execution (countdown sound, screen information, timekeeping, etc.). (b) Optimal delay being computed using the simulation model based on randomly generated participants' positions. The control panel also had the possibility to generate random positions to emulate a condition in which occupants' number is known but not their correct positions.

more than three times in the last five frames. This condition ensured that people could be accurately detected in real-time, considering that sensors had a frame rate of 30 frames per second (fps).

Overall, considering all the experiments in which the sensors have been used, sensing accuracy was of 96.9%, meaning that in a configuration with 18 people per room (which was the most typical set-up when employing the sensors) normally at most one person would not get detected.

Finally, a control panel as seen in figure 16 has been designed to control every aspect of the experimental execution. In the control panel, detected positions were reproduced and could be used to compute the optimal delay using the simulation model and the method discussed earlier. Alternatively, delay could be set manually for each room. The control panel also managed the transition from the 'in preparation', 'wait' and 'leave' indications and allowed to show the current experimental time on the main camera (the screen seen in the centre of the room in figure 13a) which was later used to synchronize videos (and thus extracted trajectories) with experimental time.

The control panel also allowed to generate random positions based on a manually provided number of participants. This option allowed to consider the scenario in which occupants' number in each room is known, but their exact positions are not (figure 16b).

Once relevant information had been provided and optimal egress time eventually computed, experiments were initiated from the control panel, which would take care of the countdown and any

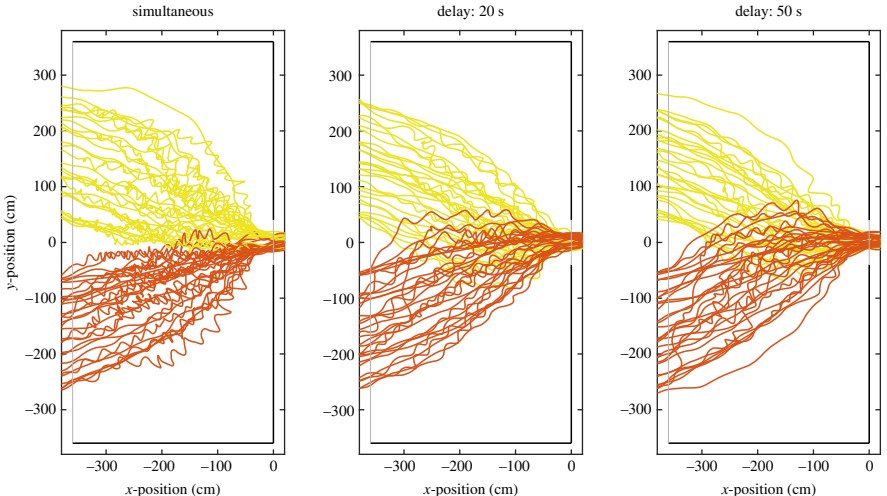

**Figure 17.** Example of trajectories for experiments in which both groups started with different delays. Different colours were used to distinguish between people coming from both waiting areas. Physical borders are given in black with the boundaries of the reference area given in grey. Exit position is set at $x, y = (0, 0)$.

other aspect related with timing. For each experiment, detected position and settings used in the control panel were stored to later assess sensing accuracy or mistakes in the procedures.

## 4.3. Participants

In total, 62 participants were recruited for the experiment and received a financial remuneration for their participation. Among them, 32 participants were university students mostly in their 20s and another 30 participants were recruited through the help of an association promoting active participation of elderly in the society and were therefore in their 60s or older. All participants were physically fit and voluntarily applied for their participation.

Participants received information in regard to privacy and data handling but were not informed on the scope or the goals of the experiments. Recruiting information simply stated that they would need to walk in partially crowded conditions.

# 5. Results and discussion

In this section, we will present and discuss the main findings of this work. All results were obtained by analysing the trajectories of pedestrians extracted from a camera mounted in azimuthal position roughly above the exit position and fixed at a height of around 6 m (figure 13a shows an example of a video frame). Camera resolution was set at $1920 \times 1440$ pixels and a frame rate of 30 fps was used.[5]

Trajectories were extracted from the video using PeTrack software [41,42] based on participants' hat colour which was used as marker. PeTrack software allows to fix both camera distortion and also account for the three-dimensional nature of people's silhouette (head position corresponds to feet position right below the camera, but both features are not aligned in external regions). Consequently, obtained trajectories are relative to the planar motion of each individual.

It should be remarked that, although people could be actually seen even when waiting inside the room, trajectories were only extracted for a region slightly larger than the reference area. Therefore, participants were tracked from slightly before entering the reference room until slightly after leaving the exit. Extracted trajectories were synchronized with the experimental time using a display shown in the middle of the room and clearly visible from the camera.

Figure 17 presents some trajectories for different experiments. Although trajectories alone do not provide much information on people's behaviour, some characteristic features can be already seen in figure 17. In particular, it is seen that when a delay is set between groups, trajectories from both

---

[5]The camera itself is not part of the system proposed in this work and its scope is solely to evaluate the performance of the system. The reason for choosing a camera to get pedestrian trajectories (instead of LiDAR sensors) is that it is more accurate, it could be placed over the room more easily and obtained videos also allow a visual inspection of the experiments.

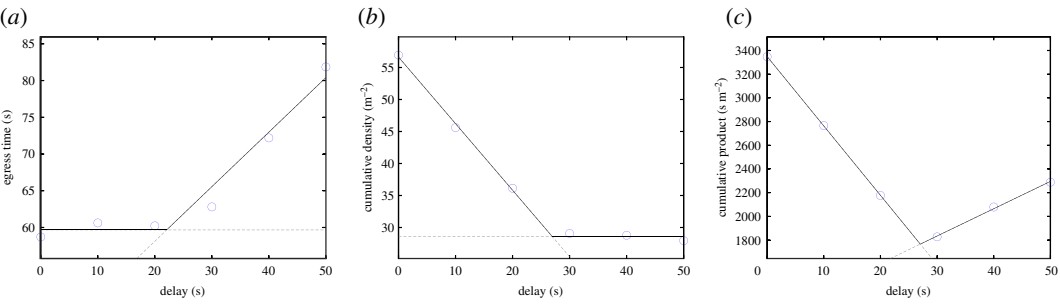

**Figure 18.** Egress time (*a*), cumulative (experienced) density (*b*) and the product of both factors (*c*) relative to the imposed delay. An optimal delay is also obtained in the experimental case.

**Table 2.** Experimental conditions and number of trials in the model validation experiments. In one trial relative to the 50 s condition one participant transited to the exit twice (after returning to the waiting room), thus having a single trial with 61 people. This issue had little or no influence on the results (especially considering the optimal delay calculation).

| | participants total number | | |
| start delay | Area A | Area B | trials |
| --- | --- | --- | --- |
| 0 s (simultaneous) | 30 | 30 | 4 |
| 10 s (A or B first) | 30 | 30 | 3 |
| 20 s (A or B first) | 30 | 30 | 3 |
| 30 s (A or B first) | 30 | 30 | 3 |
| 40 s (A or B first) | 30 | 30 | 3 |
| 50 s (A or B first) | 30 | 30 | 4 |

waiting areas overlap, whereas a clear division exists for simultaneous start. This shows that, in a delayed start, people are able to efficiently use the whole room, while in the simultaneous case space limitations are imposed by the presence of a large number of people. Also, strong swaying is seen for the simultaneous start, indicating that people are moving slowly or partially waiting. When the 20 and 50 s cases are compared, differences are minimal, but we should also remind that the temporal variable is not taken into account in the trajectories of figure 17.

## 5.1. Simulation model validation

In the first set of experiments, we checked whether the method proposed to compute the optimal delay would hold true also for real conditions and whether the simulation model was able to compute the optimal delay with sufficient accuracy. In total, 60 participants took part in this experiment and were equally divided into both rooms. In each experiment, participants occupied the front positions, thus filling the first five rows.

Start delay was manually set between 0 and 50 s in steps of 10 s and egress order was randomly chosen for each trial (either A–B or B–A). At least three repetitions were performed for each condition as summarized in table 2.

For each trial, egress time, cumulative density and their product were computed using exactly the same criteria employed in simulation. Reference area had the same size, and cumulative density was normalized to account for 1 s steps and ensure that results from experiment and simulation are comparable despite the different time steps. Results for the three quantities in the experimental case are shown in figure 18.

Generally, a satisfactory agreement is found with numerical simulation results, resulting in an optimal delay of 26.93 s (slightly higher than in simulation). The higher time seen in experiments could be partially explained with the speed set in simulations, which was about 20% higher than the one observed in experiments. However, despite a partial disagreement in quantitative terms, the 'V' shape is qualitatively confirmed as a method to compute optimal delay, although in the experimental case

**Table 3.** Experimental conditions used to test the overall system. In all cases, participants were able to occupy any of the 42 positions available in the waiting room. Strictly speaking, the optimal delay was computed also in the 0/36 condition, but obviously resulted in the occupied room leaving at the start signal, thus showing that, while not necessary, the system would work also in this case. Note: due to a mistake during the experiments, in the 18/18 case, simultaneous start experiments were taken from the asymmetric experiments described in the next section (two experiments were randomly chosen out of four). Thus, in this condition participants were not randomly distributed within the starting area, but we found this fact irrelevant (as also discussed later).

| condition | participants number | | egress start | trials |
| --- | --- | --- | --- | --- |
| | Room A | Room B | | |
| 0/36 | 0 | 36 | reference scenario | 4 |
| 18/18 | 18 | 18 | simultaneous | 2 |
| 12/24 | 12 | 24 | simultaneous | 2 |
| 6/30 | 6 | 30 | simultaneous | 2 |
| 18/18 | 18 | 18 | optimal delay computed | 4 |
| 12/24 | 12 | 24 | optimal delay computed | 4 |
| 6/30 | 6 | 30 | optimal delay computed | 4 |

the increase for large delays is less marked (slope in the right side is less marked compared with simulation).

## 5.2. System performance check

Having confirmed the capability of the simulation model in computing the optimal delay, we now wish to check whether the overall concept works in a real small-scale environment. For this purpose, the system presented earlier has been tested in a variety of scenarios. In this set of experiments 36 participants took part and were allowed to occupy any position available inside the waiting room, thus resulting in different starting positions for each trial. In total, four configurations were tested:

— all participants (36) in one waiting room (either A or B)
— participants equally split: 18 and 18
— 12 participants in one room and 24 in the other
— 6 participants in one room and 30 in the other.

For each condition in which participants occupied both rooms, four trials were performed using the system: computing the optimal delay using the detected occupied positions as simulation input. Same as in the numerical analysis presented earlier, delay in simulation was increased in 2 s steps from 0 to 50 s and the optimal delay was determined taking the minimum of the cumulative product (10 repetitions were performed for each condition).[6] Optimal delay calculation took a few seconds on a standard PC without process parallelization.

Two trials were additionally performed for each condition by having participants leaving simultaneously from each room. The case in which all participants start from one room will be taken as reference, since there is no advantage nor necessity in employing the system in this case. All experimental conditions used to test the system are summarized in table 3.

For each of the three conditions the average egress time (for the whole group) and the maximum density (relative to the whole reference area) were computed and are presented in table 4 along with the LOS associated with the maximum density. It is found that when the optimal delay is computed using the system, egress time is around 5% higher compared with the simultaneous start. However, the maximum density is reduced by almost 35%, resulting in a two levels reduction in the LOS representation. It is therefore confirmed that the system proposed in this work could actually help reducing density while having a limited effect on the egress time.

---

[6]In the presentation of the results a linear fitting has been performed on both sides of the 'V', but in the experiments simply the minimum cumulative product was taken as the optimal delay.

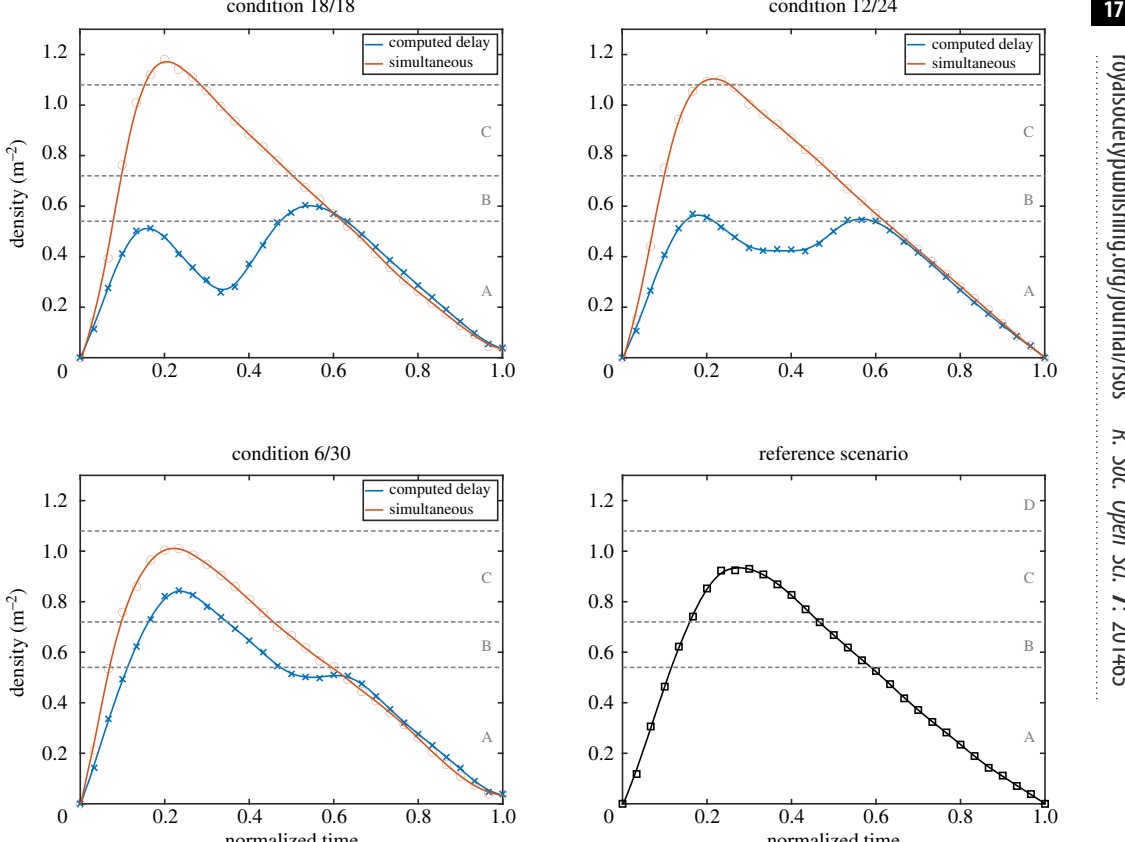

**Figure 19.** Average density profile in the different conditions and depending on the egress strategy employed. To allow a better comparison, normalized egress time is used, in which all experiments finish at 1 when the last participant transits through the exit.

**Table 4.** Egress time, maximum density and relative LOS for the three scenarios considered. All conditions were grouped in computing the average and the standard deviation.

| egress strategy | trials | egress time | maximum density | LOS |
|---|---|---|---|---|
| reference scenario | 4 | $36.97 \pm 0.57$ s | $0.947 \pm 0.020$ m$^{-2}$ | C |
| simultaneous | 6 | $36.45 \pm 1.52$ s | $1.099 \pm 0.083$ m$^{-2}$ | D |
| optimal delay computed | 12 | $38.35 \pm 0.56$ s | $0.715 \pm 0.107$ m$^{-2}$ | B |

To understand how density is reduced and what is the difference between the simultaneous and scheduled egress strategy, density profile for all conditions has been computed. To allow a better comparison and owing to the fact that egress times were not much different in the three scenarios, each trial has been divided in 30 stages using the egress time, and the average density in each stage was computed. Later, equivalent conditions were grouped and a spline fit was used to better visualize the results which are presented in figure 19.

Results show that all the cases in which a simultaneous start was imposed are almost identical and equivalent to the reference case. The maximum density is higher in the 18/18 case as people from both waiting areas quickly fill the reference area, while in the 0/36 case participants tend to line up leaving some people behind in the waiting room. When the optimal delay was employed it is seen that its efficacy is related to the distribution of people in both rooms. In the 6/30 condition the system has a limited capacity in reducing maximum density, because a single room already accommodates the majority of the participants. Quite surprisingly, the system performed better in the 12/24 condition, but this could be partially due to human factors (people's behaviour may slightly change in an unpredictable way) or due to the limited accuracy of the simulation model (remember that there is a difference of about 6 s between experiments and simulation in predicting the optimal delay). In both

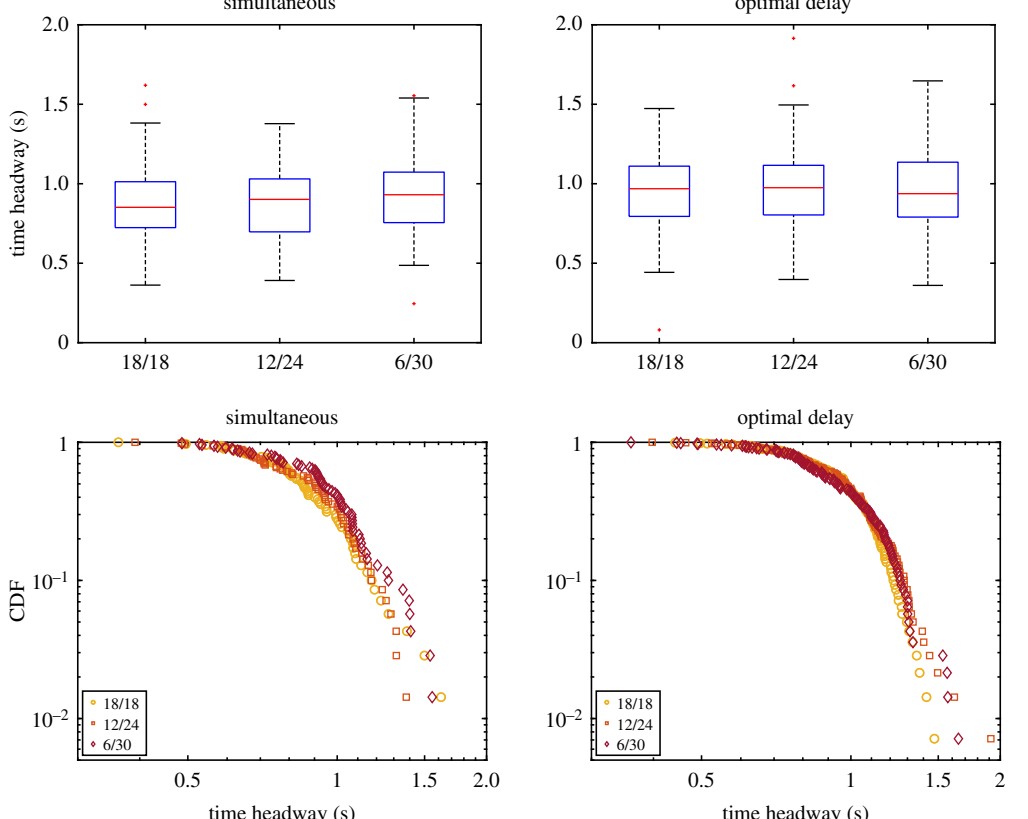

**Figure 20.** Temporal headway distributions for the different conditions and depending on the egress strategy employed. In the plots on the top, the central mark stands for the median, with the bottom and top edges indicating the 25th and 75th percentiles, respectively. Black lines are used for minimum and maximum, with outliers indicated as red crosses.

12/24 and 18/18 cases, it is nonetheless clear that density is greatly reduced because distributions are optimally shifted in time.

We now wish to investigate a little further the differences between the various egress strategies and whether there is a difference between conditions in relation to room occupancy. Figure 20 presents the headway at the exit for each condition in the case of simultaneous or delayed egress. The (time) headway is defined as the time difference between consecutive participants passing through the exit and is often used to assess clogging or crowding conditions. Both the statistical plot and the cumulative distribution function (CDF) representations of figure 20 show that when the egress process is analysed at the exit, all conditions are equivalent also regardless of the egress strategy employed.

In the previous approach, time headway was computed between individuals, regardless of their starting room. But, as discussed in the introduction, there may be situations in which it would be necessary to avoid mixing groups (consider the COVID-19 pandemics or soccer fans of opposite teams). We now wish to consider the time headway between successive participants belonging to the same group. From this perspective, headway tends to get larger as groups mix with each other.

Results for the group-specific (or in-group) headway are presented in figure 21a,b. This time, we will separately consider the first and the second group to leave, regardless of the waiting room from which they started. As seen in the CDF plots, long headways are much more common in the case of simultaneous start, with headways longer than 1.5 s accounting for more than 30% of the total. On the other side, when optimal egress delay was computed, headways longer than 1.5 s account for less than 5% of the total. This confirms the fact that groups are more compact and less mixed in scheduled egresses.

To conclude the section, we now wish to consider the group time separation, i.e. the difference in time between the last person of the first (leading) group and the first person of the second (following) group. The results presented in figure 21c clearly show that when the delay was computed, time between groups was of a few seconds and almost equally distributed around zero (perfect timing, i.e. one group perfectly following the next one). On the other side, the simultaneous case always resulted in big negative times, an indication that groups were completely mixed.

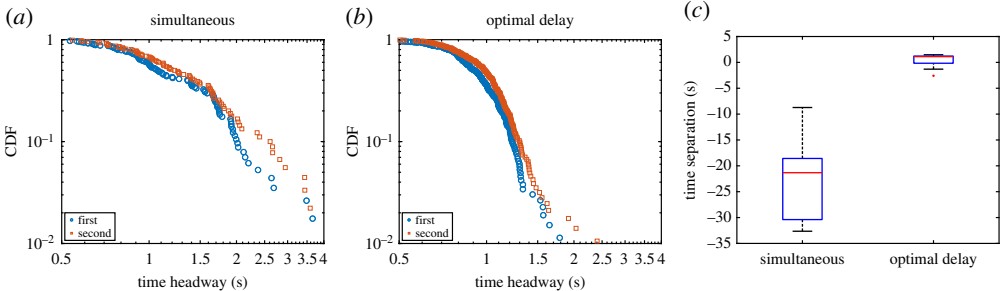

**Figure 21.** (*a,b*) In-group headway CDF depending on the egress strategy employed. All conditions are grouped together, but, here, the reference scenario is excluded since it consists of one group only. (*c*) Time difference between the last person of the first group and first person of the second group for each strategy.

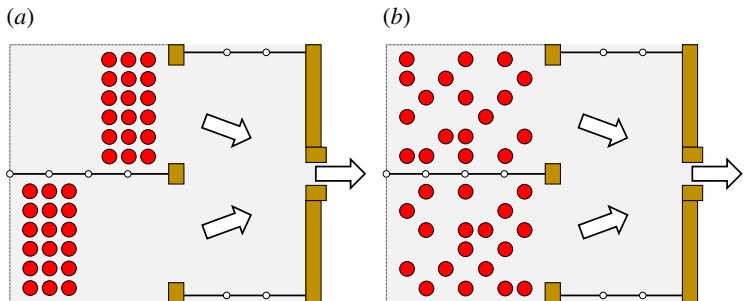

**Figure 22.** (*a*) Participants' real starting position in the asymmetric experiments. In the set-up represented here participants in room A (top) take front positions, but also the opposite set-up (i.e. the case in which participants in room B took front positions) was tested. (*b*) Example of randomly generated positions used in simulation when only the total number of occupants was used.

## 5.3. Asymmetric start experiments

In the previous section, we showed that when the combination of sensors-simulation was used to compute the optimal delay, a less crowded and well-divided egress process could be achieved by slightly increasing the overall egress time. It is, however, questionable whether, in clearly divided spaces, the accurate position of people is necessary information or if the total number of occupants would be sufficient.

To investigate this problem, an additional set of experiments was performed using a feature of the experiment control panel described earlier. Participants (again 36 in total in this set of experiments) were divided into two equal groups and were asked to occupy either the 18 front positions or the 18 back positions in each waiting room, thus resulting in a configuration like the one presented in figure 22*a*. In this set of experiments (labelled 'asymmetric') participants always started in a front–back configuration, but three strategies were used in regard to egress scheduling:

— Location data: participants' starting positions were obtained through the sensors and were used in simulations to compute the optimal delay.
— Total only: assuming only the total number of participants is known, participants' positions were randomly generated (see the example in figure 22*b*) and these randomly generated positions were used in simulation to compute the optimal delay. Participants number (18) was manually given in the control panel assuming the total number of people is counted at the entrance (for example, using transit gates).
— Simultaneous: participants from both rooms started simultaneously.

For each experimental condition, four trials were performed in total, with groups A and B starting in the front positions two times each. Experimental conditions are also summarized in table 5.

An initial analysis based on egress time and density profiles (presented in figure 23) revealed that, while the use of the system can help reduce density by slightly increasing egress time (as already confirmed with the previous experiments), we did not find any significant difference between the approach based on accurate positions and the one only relying on occupants' total number. Both

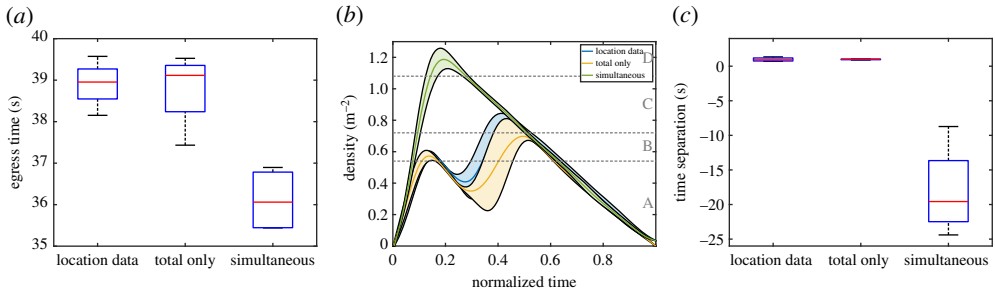

**Figure 23.** Egress time (*a*), density profile (*b*) and time separation between groups (*c*) for the three experimental conditions considered here. In (*b*) error range (light background) is considered in the density plots along with the average line. While a clear difference is seen when the system is employed, differences are minimal and statistically insignificant between the approach based on participants' position and the one based on total number.

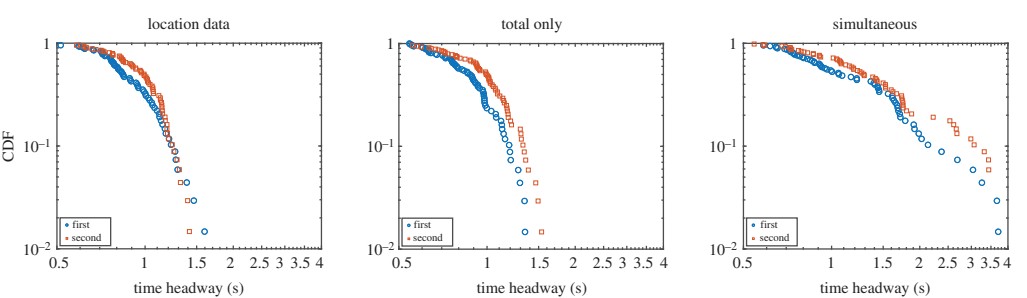

**Figure 24.** In-group headway CDF for the three conditions considered in the asymmetric set of experiments. First and second stand for the first group to exit, regardless on their starting room.

**Table 5.** Conditions tested in the asymmetric experiments, number of participants per room and number of trials.

| experimental condition | participants number | | trials |
| --- | --- | --- | --- |
| | Room A | Room B | |
| accurate location data | 18 | 18 | 2 (A front) and 2 (B front) |
| occupants' total number only | 18 | 18 | 2 (A front) and 2 (B front) |
| simultaneous | 18 | 18 | 2 (A front) and 2 (B front) |

differences in terms of egress time and density between the location-based and total-based approach are minimal and insignificant from a statistical point of view. A minimal, but possibly interesting difference, lies in the egress time, where it is found that the use of the individuals' positions tends to produce more stable results (distribution is more compact) possibly because optimal delay is computed more accurately and systematically.

Also, when the time separation between groups is considered (figure 23*c*), the validity of the system is confirmed, but no particular merit is seen in obtaining accurate waiting positions. Possibly also owing to the fact that all configurations were equal in occupants per room, a very small variation in separation time is seen for both cases in which the system was employed, confirming the capability to lead to stable results under different conditions.

Hoping to find some clue in the distribution of the in-group headway (i.e. the headway of people belonging to the same waiting area), the CDFs for the three conditions were computed and are presented in figure 24. Also in this case, the same main conclusions found for the previous results hold true: there is a clear difference between scheduled and simultaneous egress, but differences between both simulation-based approaches are minimal. In contrast to the previous set of experiments (system performance check) a slightly larger difference is seen between the first and second groups, possibly related to the asymmetric starting configuration. Similar to the egress time considered earlier, also in the graphs of figure 24, the position-based approach reveals a slightly more rapid drop in the distribution of headway, possibly indicating a more precise scheduling, but, again, differences are minimal and largely insignificant.

It can be therefore concluded that, at least for a scenario like the one considered here, relying on the overall number of people per section (or room) may be sufficient. However, for large complex structures where occupants' positions may take very different configurations, having access to precise information may be useful, although probably not strictly necessary.

## 6. Conclusion

In this work, we presented a method and a system which can be used to schedule visitors' egress timing in complex structures under normal conditions with the aim to reduce density while also avoiding a mixture between different groups of people. The concept was first presented schematically employing different case studies and later tested experimentally using an ad hoc system specifically designed to test its potentials and limitations in a small-scale scenario.

Experimental results clearly show that when egress timing is scheduled based on information gained in regard to occupants' total number (or attendance), a strong reduction in density was observed, although total egress time was only slightly increased. Also, we found that, at least for the simple scenario tested here, the total number of occupants could be sufficient in performing the calculations for the optimal delay(s), with actual positions possibly helping in increasing the accuracy of prediction only marginally.

The encouraging results show that it is possible to increase visitors' overall satisfaction in conditions where critical congestion is not an issue using some relatively simple equipment or information already available to facility operators (such as the number of people in each section based on ticket sales). In extremely crowded conditions, like in a train station terminal where trains are constantly arriving, it is, however, questionable whether such a system would be beneficial as delay imposed (for example, on train arrival) may slow down overall operations and possibly make conditions even worst.

Thus, while we do believe that such a system can be useful in many situations, more research is needed to assess the trade-off between the relevance of comfort and time loss from the perspective of pedestrian users. In a situation like a multiplex cinema or a conference venue, delays could be imposed without users noticing it by adjusting events' schedule. But in many other situations, visitors may be more interested in their time rather than being concerned about congestion.

To conclude, we also need to acknowledge limitations in our work, namely considering the limited size of the experimental setting and the fact that participants were recruited individuals acting according to specific instructions. In real conditions, people may not leave in group even from the same area/room, thus making simulation and consequently egress scheduling more challenging and potentially useless from a practical perspective.

Ethics. The experiment was approved by the Ethical Commission of The University of Tokyo and conforms with the Declaration of Helsinki.

Data accessibility. Data relative to the experiments discussed in this work are provided along this manuscript in form of electronic supplementary material.

Authors' contributions. H.M. and C.F. conceived the general idea and planned the experiments. C.F. developed and built the technical system and prepared the electronic supplementary material, and H.M. wrote the simulation program. C.F. extracted trajectories from the videos based on which C.F. and H.M. computed the results. C.F. and H.M. wrote the manuscript which was later agreed by all authors. The theoretical introduction was prepared by C.F. based on ideas from all authors. K.N. supervised the project and obtained the necessary funds.

Competing interests. We declare we have no competing interests.

Funding. This work was financially supported by the JST-Mirai Program (grant nos. JPMJMI17D4 and JPMJMI20D1) and the JSPS KAKENHI (grant nos. JP20K14992 and JP20K20143).

Acknowledgements. The experiments were organized with the help of the members of the Nishinari laboratory. In addition, the authors would like to thank the Meguro Silver Human Resource Center for their participation in the experiments.

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
