## [Reviewer comments · Royal Society Open Science]

Review History

RSOS-201465.R0 (Original submission)

Review form: Reviewer 1

Is the manuscript scientifically sound in its present form?

Yes

Are the interpretations and conclusions justified by the results?

Yes

Is the language acceptable?

Yes

Do you have any ethical concerns with this paper?

No

Have you any concerns about statistical analyses in this paper?

No

Recommendation?

Accept with minor revision (please list in comments)

Comments to the Author(s)

Crowd management under critical and non-critical conditions is a relevant topic addressed in this paper by proposing an adaptive egress scheduling system. The scheduling is based on finding the optimal delay between the egress time of different pedestrian groups. The paper is well structured and written, with a theoretical and conceptual background presented in a very illustrative way. I have the following questions.

The idea of finding the optimal delay between the egress time of different pedestrian groups is not new. For instance, in Abdelghany, A., Abdelghany, K., Mahmassani, H., & Alhalabi, W. (2014). Modeling framework for optimal evacuation of large-scale crowded pedestrian facilities. *European Journal of Operational Research*, 237, 1105–1118. doi:10.1016/j.ejor.2014.02.054., the optimal delay for each region of a facility is found through a simulation-optimization approach using a genetic algorithm. They need to apply a non-linear optimization algorithm because the search space is not trivial, including several exit gates, delays, and many regions. The only difference between these works regarding the conceptual framework seems to be in the optimization objective. In Abdelghany's paper, the goal is to optimize the evacuation time. However, your goal is to optimize both evacuation time and density. I would ask the authors to state which is their main contribution regarding the conceptual framework presented in the paper.

In the Simulation model (FF), it would be convenient to indicate the S_{ij} values for reproducibility. Are they proportional to distance and scaled to 1?

For this reviewer, the strength of this paper is in the validation of the approach. The experimental design has a high quality, and it is convincing. Regarding the technical setup, I do not understand why you have used two different systems: LIDAR sensors to detect pedestrian locations and a video camera to track trajectories. Would it be possible to use the video camera to perform both tasks? Which is the advantage of using both systems?

Considering that the idea of optimizing the delay is not new, it should be clearly detailed, which are the contributions of the article.

I have found the following mistakes in the text:

- 1) Abstract: " groups of people living ..."
- 2) Pg.8 L.57 Figure 1 should be Figure 7.
- 3) Pg.17. L.18 Figure 12 should be Figure 13.
- 4) Pg.18. L.14 60 participants ... should not start with a number.
- 5) Pg.19. In Table 4 the reference scenario ... should not be LOS C??
- 6) In Figure 22, review "take front positions, ...tested"
- 7) Pg.23 L30 "There a clear"

Review form: Reviewer 2

Is the manuscript scientifically sound in its present form?

Yes

Are the interpretations and conclusions justified by the results?

Yes

Is the language acceptable?

Yes

Do you have any ethical concerns with this paper?

No

Have you any concerns about statistical analyses in this paper?

No

Recommendation?

Accept with minor revision (please list in comments)

Comments to the Author(s)

In this paper, the authors proposed a system aimed at computing optimal egress time for groups of people in a complex facility. Simulation and experiment were conducted to validate the system. For me, the idea is interesting and it is acceptable after minor revision. Minor remarks:

1. In each figure, the authors need to use (a), (b), (c) to distinguish different sub figures.
2. The introduction part can be improved and the current description is not logic. In addition, the motivation and contribution should be more clear.

Decision letter (RSOS-201465.R0)

Dear Dr Feliciani

On behalf of the Editors, we are pleased to inform you that your Manuscript RSOS-201465 "A system for efficient egress scheduling during mass events and small-scale experimental demonstration" has been accepted for publication in Royal Society Open Science subject to minor revision in accordance with the referees' reports. Please find the referees' comments along with any feedback from the Editors below my signature.

Please submit your revised manuscript and required files (see below) no later than 7 days from today's (ie 15-Oct-2020) date. Note: the ScholarOne system will 'lock' if submission of the revision is attempted 7 or more days after the deadline. If you do not think you will be able to meet this deadline please contact the editorial office immediately.

Kind regards,
Anita Kristiansen

Editorial Coordinator

on behalf of Dr Jose Carrillo (Associate Editor) and Mark Chaplain (Subject Editor)
 openscience@royalsociety.org

Reviewer comments to Author:

Reviewer: 1

Comments to the Author(s)

Crowd management under critical and non-critical conditions is a relevant topic addressed in this paper by proposing an adaptive egress scheduling system. The scheduling is based on finding the optimal delay between the egress time of different pedestrian groups. The paper is well structured and written, with a theoretical and conceptual background presented in a very illustrative way. I have the following questions.

The idea of finding the optimal delay between the egress time of different pedestrian groups is not new. For instance, in Abdelghany, A., Abdelghany, K., Mahmassani, H., & Alhalabi, W. (2014). Modeling framework for optimal evacuation of large-scale crowded pedestrian facilities. *European Journal of Operational Research*, 237, 1105–1118.

doi:10.1016/j.ejor.2014.02.054., the optimal delay for each region of a facility is found through a simulation-optimization approach using a genetic algorithm. They need to apply a non-linear optimization algorithm because the search space is not trivial, including several exit gates, delays, and many regions. The only difference between these works regarding the conceptual framework seems to be in the optimization objective. In Abdelghany's paper, the goal is to optimize the evacuation time. However, your goal is to optimize both evacuation time and density. I would ask the authors to state which is their main contribution regarding the conceptual framework presented in the paper.

In the Simulation model (FF), it would be convenient to indicate the S_{ij} values for reproducibility. Are they proportional to distance and scaled to 1?

For this reviewer, the strength of this paper is in the validation of the approach. The experimental design has a high quality, and it is convincing. Regarding the technical setup, I do not understand why you have used two different systems: LIDAR sensors to detect pedestrian locations and a video camera to track trajectories. Would it be possible to use the video camera to perform both tasks? Which is the advantage of using both systems?

Considering that the idea of optimizing the delay is not new, it should be clearly detailed, which are the contributions of the article.

I have found the following mistakes in the text:

- 1) Abstract: " groups of people living ..."
- 2) Pg.8 L.57 Figure 1 should be Figure 7.
- 3) Pg.17. L.18 Figure 12 should be Figure 13.
- 4) Pg.18. L.14 60 participants ... should not start with a number.
- 5) Pg.19. In Table 4 the reference scenario ... should not be LOS C??
- 6) In Figure 22, review "take front positions, ...tested"
- 7) Pg.23 L30 "There a clear"

Reviewer: 2

Comments to the Author(s)

In this paper, the authors proposed a system aimed at computing optimal egress time for groups of people in a complex facility. Simulation and experiment were conducted to validate the system. For me, the idea is interesting and it is acceptable after minor revision. Minor remarks:

1. In each figure, the authors need to use (a), (b), (c) to distinguish different sub figures.
2. The introduction part can be improved and the current description is not logic. In addition, the motivation and contribution should be more clear.

===PREPARING YOUR MANUSCRIPT===

===PREPARING YOUR REVISION IN SCHOLARONE===

Author's Response to Decision Letter for (RSOS-201465.R0)

See Appendix A.

Decision letter (RSOS-201465.R1)

Dear Dr Feliciani,

It is a pleasure to accept your manuscript entitled "A system for efficient egress scheduling during mass events and small-scale experimental demonstration" in its current form for publication in Royal Society Open Science. The comments of the reviewer(s) who reviewed your manuscript are included at the foot of this letter.

on behalf of Dr Jose Carrillo (Associate Editor) and Mark Chaplain (Subject Editor)
openscience@royalsociety.org

Appendix A

Reviewer #1:

Crowd management under critical and non-critical conditions is a relevant topic addressed in this paper by proposing an adaptive egress scheduling system. The scheduling is based on finding the optimal delay between the egress time of different pedestrian groups. The paper is well structured and written, with a theoretical and conceptual background presented in a very illustrative way.

Thank you for your encouraging comment. We were a little afraid the theoretical and conceptual part made the paper unnecessarily long, but we are glad to see that the external reviewer judged positively our decision to provide a theoretical foundation to our concept.

The idea of finding the optimal delay between the egress time of different pedestrian groups is not new. For instance, in Abdelghany et al., the optimal delay for each region of a facility is found through a simulation-optimization approach using a genetic algorithm. They need to apply a non-linear optimization algorithm because the search space is not trivial, including several exit gates, delays, and many regions. The only difference between these works regarding the conceptual framework seems to be in the optimization objective.

Thank you for reporting us this interesting work, we added it along to another work we already knew (Guo 2018) but, at the time of writing the initial manuscript, did not consider relevant due to the fact that evacuation time optimization was its main and only objective. Now that we have rewritten the introduction to better frame the position of our work within the literature, both Abdelghany et al. 2014 and Guo 2018 fit well with the discussion and are well suited for being cited in our work.

In Abdelghany's paper, the goal is to optimize the evacuation time. However, your goal is to optimize both evacuation time and density. I would ask the authors to state which is their main contribution regarding the conceptual framework presented in the paper.

Also considering the comments of reviewer #1, we had to acknowledge that the goal and main contribution of this work were not clear. We have completely rewritten the second part of the introduction where the motivation and the particularity of our framework are presented and its contribution explained.

To summarize, the main contribution of our framework is that it provides a method to reduce crowd intrinsic risks while improving the comfort of pedestrian users. While most of the methods proposed to employ real-time pedestrian positions aims at making evacuations more efficient (i.e. faster) in case of accidents, we focus on normal operation and propose an alternative framework which could be used to improve pedestrian safety and comfort in absence of external threats (like a fire). In this sense, our framework should not be seen as a substitute to methods aiming at reducing evacuation time in accidents, but a system to be used during normal (daily) operation.

In a more practical example, the proposed system could be employed in a stadium to reduce densities at the exit and allow visitors having a more enjoyable experience (not having to move through a packed crowd after the game). However, should a fire or an earthquake occur, then the methods such as the ones proposed by Abdelghany et al., Guo or Lopez would be needed to ensure people leave the stadium as quickly as possible, also, or possibly considering the location of the threat (not considered at the moment but potentially being an interesting topic for future research).

To avoid taking too much space here we avoided reproducing the introduction, but you may check the modifications in the PDF file showing the changes from the original manuscript.

In the Simulation model (FF), it would be convenient to indicate the S_{ij} values for reproducibility. Are they proportional to distance and scaled to 1?

Thank you for the suggestion, but we prefer to avoid adding numbers inside the figure as they would become difficult to read. We preferred adding a statement in the text clarifying the values of the static FF (the added text is also reported below).

To answer your question: no, they are not scaled to 1 and are simply the minimal distance to the exit by moving in a Neumann neighborhood. This means the static FF value in each cell is the minimum distance it takes to the exit if you move only to the sides (i.e. diagonal motion not allowed).

“The static FF employed for the double room scenario is very simple (see Figure 10) and it simply represents the minimum distance (in cells) to the exit if a Neumann neighborhood is employed (i.e. in the model people cannot move diagonally but only along the four directions). Under this condition, the value of the static FF decreases from the back of the rooms (where it equals 20 “steps”) until reaching its minimum (zero) at the exit.”

For this reviewer, the strength of this paper is in the validation of the approach. The experimental design has a high quality, and it is convincing.

Thank you again for your positive feedback and for motivating us in continuing doing research in this direction.

Regarding the technical setup, I do not understand why you have used two different systems: LIDAR sensors to detect pedestrian locations and a video camera to track trajectories. Would it be possible to use the video camera to perform both tasks? Which is the advantage of using both systems?

This is certainly a valid comment which would need more clarifications. As a matter of fact, the camera is not properly part of the system itself, but only used to evaluate the performance of the system in the experiments. In a practical application there is no need to check the validity of the system as this has been already confirmed here (of course it would be reasonable and interesting to further evaluate the system, but it is not strictly necessary).

As the reviewer commented, in fact, we could have used only cameras or only Lidars. The reason for choosing both is that we are envisaging the use of Lidar for real applications as they are easier to accept from facility operators since privacy is not a concern. The reason for using the camera is related to practical reasons. The only way to get a top view is by employing a lightweight camera which we can fix on the light supports on the ceiling as seen in the image below (note: in this image the camera is not secured, in the experiments it has been locked using screw to avoid falling even in case of earthquakes). Also, there is a range of software which can be used to extract trajectories from videos and generally speaking, the quality of trajectories is better compared to Lidar. In this sense, we favored quality over practical applications in regard to collection of trajectories as these are only used to evaluate the performance of the system.

In any case, to avoid misunderstandings and make those aspects clearer to the readers we revised the text with the relevant parts provided below the image in orange color.

“The camera itself is not part of the system proposed in this work and its scope is solely to evaluate the performance of the system. The reason for choosing a camera to get pedestrian trajectories (instead of LiDAR sensors) is that it is more accurate, it could be placed over the room and obtained videos also allow a visual inspection of the experiments.”

Considering that the idea of optimizing the delay is not new, it should be clearly detailed, which are the contributions of the article.

Correct; we answered this point while answering your previous concern in which you suggested us the work by Abdelghany et al. We believe we already answered this point in the comments provided beforehand and we hope at this point the contribution should be clear or at least clearer than in the original submitted manuscript.

I have found the following mistakes in the text:

- 1) Abstract: " groups of people living ..."
- 2) Pg.8 L.57 Figure 1 should be Figure 7.
- 3) Pg.17. L.18 Figure 12 should be Figure 13.
- 4) Pg.18. L.14 60 participants ... should not start with a number.
- 5) Pg.19. In Table 4 the reference scenario ... should not be LOS C??
- 6) In Figure 22, review "take front positions, ...tested"
- 7) Pg.23 L30 "There a clear"?

Thank you for carefully checking the manuscript and pointing out these points. We have performed the necessary modifications in the text (all of them were appropriate and employed except the one concerning the abstract which has been rewritten to makes it shorter).

Reviewer #2:

In this paper, the authors proposed a system aimed at computing optimal egress time for groups of people in a complex facility. Simulation and experiment were conducted to validate the system. For me, the idea is interesting and it is acceptable after minor revision.

Thank you for the encouraging evaluation of our work. We are also glad to see that the most relevant points listed from the reviewer correspond to what we see as the strengths of our work, giving us a hint that the manuscript managed to deliver the right message.

In each figure, the authors need to use (a), (b), (c) to distinguish different sub figures.

Thank you for pointing this out. We added (a), (b), (c) to the figures which were described using “Left”, “Right” or so on, but we decided not to add the letters to the images which are self-explaining (especially the ones having a clear title above the graphs). We believe that adding additional information would make their interpretation more difficult instead of making it easier. In any case, if adding the letters to every single image is necessary due to the journal policy, we will add the letters during a later revision or proof-reading.

We also did a short research on papers recently published in Royal Society Open Science and it seems that letters can be omitted if the content is clear by providing titles (see for example [10.1098/rsos.191983](https://doi.org/10.1098/rsos.191983) or [10.1098/rsos.191998](https://doi.org/10.1098/rsos.191998)).

The introduction part can be improved and the current description is not logic. In addition, the motivation and contribution should be more clear.

Having a fresh read of the manuscript after a couple of months, made us aware of some deficiencies in the introduction and we have to agree with the reviewer that a revision regarding that part is needed (reviewer #1 also had similar comments). We have therefore rewritten the introduction and the abstract (which was too long and not compliant with the journal guidelines).

Since modifications are distributed along the introduction itself, we avoided reproducing the whole introduction again here and we would ask the reviewer to refer to the latest manuscript (we are providing a file with all modifications in case you want to check in detail what has been changed).